# Sim-to-Real via Sim-to-Seg: End-to-end Off-road Autonomous Driving Without Real Data

**John So**[*†]   **Amber Xie**[*†]   **Sunggoo Jung**[‡]   **Jeffrey Edlund**[‡]   **Rohan Thakker**[‡]

**Ali Agha-mohammadi**[‡]   **Pieter Abbeel**[†]   **Stephen James**[†]

[†]UC Berkeley   [‡]NASA Jet Propulsion Laboratory   [*]Equal Contribution

**Abstract:** Autonomous driving is complex, requiring sophisticated 3D scene understanding, localization, mapping, and control. Rather than explicitly modelling and fusing each of these components, we instead consider an end-to-end approach via reinforcement learning (RL). However, collecting exploration driving data in the real world is impractical and dangerous. While training in simulation and deploying visual sim-to-real techniques has worked well for robot manipulation, deploying beyond controlled workspace viewpoints remains a challenge. In this paper, we address this challenge by presenting Sim2Seg, a re-imagining of RCAN [1] that crosses the visual reality gap for off-road autonomous driving, without using any real-world data. This is done by learning to translate randomized simulation images into simulated segmentation and depth maps, subsequently enabling real-world images to also be translated. This allows us to train an end-to-end RL policy in simulation, and directly deploy in the real-world. Our approach, which can be trained in 48 hours on 1 GPU, can perform equally as well as a classical perception and control stack that took thousands of engineering hours over several months to build. We hope this work motivates future end-to-end autonomous driving research. Code and videos available on our project page.

**Keywords:** Sim-to-Real, Reinforcement Learning, Autonomous Driving

## 1   Introduction

Simplifying large autonomous driving software stacks, which are usually composed of 3D scene understanding, localization, mapping, and control, is a promising goal. While these stacks can indeed perform well in a range of scenarios, they do suffer from error propagation through each of the modules, and tend to require a large engineering overhead. However, attempting to solve autonomous driving problem in a purely end-to-end manner, where observations are mapped directly to actions, also has its downfalls. For one, these methods are usually data intensive, and in particular for reinforcement learning (RL), collecting exploration driving data in the real world is impractical and dangerous.

To overcome the data burden, large-scale simulations can be employed to collect experience from a large number of parallel agents. However, we then have to consider the visual and dynamic discrepancies between the simulation and reality. Sim-to-real transfer approaches, such as domain adaptation [2, 1] and domain randomization [3, 4, 5, 6] exist for this reason. These however, have mostly been applied in tasks with a fixed camera viewpoint, such as manipulation tasks where the camera usually points down towards a bin [2, 1] or table [3], or faces a wall [4, 5].

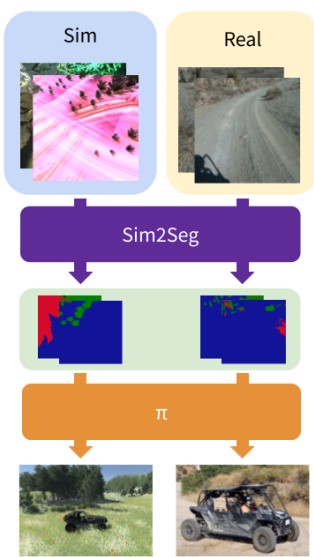

Figure 1: Sim2Seg learns a mapping between randomized images and segmentation maps for zero-shot transfer.

6th Conference on Robot Learning (CoRL 2022), Auckland, New Zealand.

Unlike these robot manipulation setups, autonomous navigation with full-scale vehicles in off-road settings are far less controlled, often featuring aspects such as drastically different lighting, glare, dust, long visual horizons, and complex backgrounds. In addition, unlike in standard autonomous street driving, the vehicle must traverse off-road terrains, which have less uniform dynamics and may include natural obstacles such as bushes, rocks, bumps, and more, which may not be observed during training.

In this paper, we investigate how similar sim-to-real techniques used for smaller scale robotic domains, such as robot manipulation, can be scaled to the significantly more visually challenging domain of off-road autonomous driving of a large vehicle. To this end, we propose Sim2Seg, a sim-to-real method designed with the challenges of off-terrain autonomous navigation in mind. Combined with a deep RL policy, Sim2Seg is the first work to effectively employ primarily visual sim-to-real transfer for off-road autonomous driving.

**Contributions**   We highlight our contributions below:

- We improve the discriminator component within RCAN [1] by convolving the output segmentation maps to produce feature maps that are then fed to a discriminator to evaluate.
- We explore suitable action modes that encourage safe trajectory proposals without diminishing the model's capability.
- We show that our trained RL policy can perform as well as a sophisticated, model-based autonomous driving stack.
- We show the first primarily visual sim-to-real transfer method for end-to-end RL autonomous driving in complex off-road terrains that requires no real world data.

## 2   Related Work

**Model-based Autonomous Driving**   Most off-road autonomous driving approaches center around scene understanding approaches. Due to the challenges of natural obstacles such as bushes, trees, and rocks; uneven terrain; and various static and dynamic uncertainties [7], the robot must constantly assess the traversability of the terrain. Geometry-based methods involve constructing a terrain map [8, 9] from depth measurements from sensors such as LiDAR, stereo cameras, etc. This terrain map is used to generate a traversability cost by performing stability analysis, using features like surface normals, maximum or minimum height of the terrain, etc., which can be used by motion planning and control algorithms to plan vehicle's actions [10, 11, 12, 13]. Papadakis [14] provides a survey of several other off-road driving algorithms. In lieu of these methods, we focus on end-to-end autonomous driving directly from pixels, avoiding these engineering layers.

**End-to-End Autonomous Driving**   End-to-end autonomous driving at large remains an unsolved problem: most approaches have narrowed their scope to urban environments [15, 16], as it is most relevant for everyday human transportation. In addition, most methods simplify the problem of general navigation by assuming static environments [17] and real-world datasets [15], which is a limiting factor for more difficult terrains and navigation tasks like off-road autonomous driving.

Since urban environments produce unique challenges of multi-agent interactions and following traffic rules, many end-to-end approaches greatly simplify the environment to focus on navigation on roads. For instance, Chu et al. [18] and Nair et al. [19] both reduce the environment to static, toy car racing environments. While Kendall et al. [20] trains a visual RL policy end-to-end in simulation, they focuses explicitly on lane-following in a static environment and requires real-world policy rollouts for few-shot policy transfer. Offline real-world data has also been crucial to many methods. Ram [21] trains end-to-end in CARLA [22] — an urban driving simulator, but to bridge the visual sim-to-real gap, requires real-world data in order to enhance simulator images. VISTA [15] focuses on street navigation and relies upon a data-driven simulator, which synthesizes new viewpoints of a scene based on offline data, making it difficult to quickly simulate new scenes. Osinski et al. [17] uses real-world images and ground-truth semantic segmentation maps to learn segmentations. We instead focus on a goal-conditioned policy for the explicit task of off-road navigation and obstacle avoidance, and zero-shot policy transfer to the real world via learning a shared representation from a diverse set of high-fidelity simulations.

**Sim-to-Real in the Visual Domain**    Perception forms the basis of many tasks, from smaller-scale robotics tasks to large-scale vehicles. To ensure consistent perception across simulators and real-world images, a popular technique used is domain randomization [23, 4, 16], in which input observations are randomized to prevent overfitting to simulator images, ensure adaptability to a variety of conditions, and encourage extraction of meaningful features such as object shapes and locations.

In the case that real-world data is easily accessible, domain adaptation is a popular method used to extract consistencies across the two domains. Pixel-level domain adaptation, for instance, improves pixel-level consistencies by restyling simulator images [24, 25]. For robotics tasks involving objects that are critical to the scene, additional losses are penalized: RetinaGAN ensures object consistencies using a pretrained object detector [26], and RL-CycleGAN [27] penalizes differences in Q-values. Feature-level domain adaptation learns shared features across both domains [28, 29, 30]. It is notable that many of these environments are based around grasping and other robotics tasks, which have fixed camera viewpoints, controlled environments, and relatively easier data collection processes in comparison with off-road vehicles.

The most direct analog of our approach is RCAN [1], which approaches visual sim-to-real translation in robotic grasping via learning a shared RGB *canonical* space using a Pix2Pix model [31]. While directly inspired by RCAN, our method has distinct differences, which we highlight: (1) RCAN converts visual inputs to canonicalized RGB images. Sim2Seg converts visual inputs to one-hot segmentation maps, which are a different, relatively lower dimensional, modality. (2) RCAN discriminators take in paired RGB images. To improve discriminator stability, our discriminator takes in learned features of input images and segmentation maps. (3) RCAN approaches robotic manipulation tasks. Sim2Seg approaches autonomous driving in the much more difficult environment of offroad environments, which presents additional challenges as described in Section 1.

## 3    Method

In this section, we detail our method, Sim2Seg, which is summarised in Figure 1. It consists of two primary components: (1) inspired by RCAN [1], we train a Sim2Seg model that translate randomized RGB images from simulation into a shared *canonical* form, which we define as a semantic segmentation map. (2) We then train a goal-reaching policy within this canonical representation to navigate in off-terrain environments. During inference, we use a frozen pretrained Sim2Seg model to perform zero-shot transfer on real-world images.

### 3.1    Simulation

We create several simulation environments using the Unity engine. Unity fulfills several key desiderata, notably high-fidelity visual observations and dynamics, open-source vehicle components and scenes [32], and the ability to apply custom domain randomization techniques. Unity also integrates well with RL training with the Unity ML-Agents toolkit, which provides a Gym interface for training [33]. For our purposes, we further modify ML-Agents to support instance-level parallelism, allowing us to train multiple agents per built executable.

To maximally train our policy to a variety of off-road environments and generate a diverse dataset of simulator data, we select 3 different simulated scenes — dubbed Meadow [34], Landscapes [35], and Canyon [36] — with semantically diverse visual environments. During training, the policy trains simultaneously on all 3 environments, leveraging the most out of the simulation training phase and ensuring adaptability to a variety of scenes.

### 3.2    Sim2Real via Sim2Seg

To bridge the visual gaps between simulators and real-world data, we use a Sim2Seg model to convert randomized image domains into our chosen canonical form, a segmentation map consisting of six classes: trees/bushes, ground, sky, rocks, road, and logs. Visualization colors can be found in Appendix B. We believe this is useful for off-road vehicles because it simplifies the unnecessary details, textures, and colors of images and identifies different types of obstacles, which gives important information on obstacles and areas to avoid (see Figure 2).

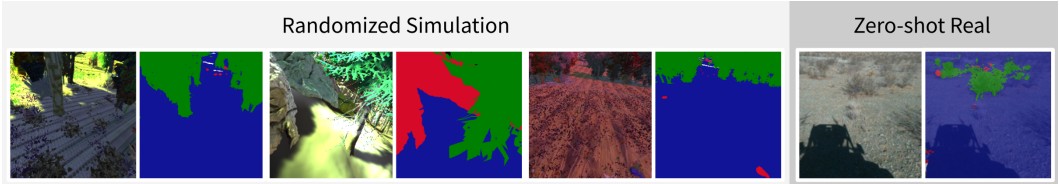

Figure 2: To bridge the visual sim-to-real gap, we apply combinations of texture randomization, camera position and intrinsic randomization, and lighting color and direction randomization to varying scenes in Unity, and learn mappings to ground truth segmentation maps.

**Domain Randomization**   For data collection, we collect 200k pairs of paired RGB images, segmentation maps, and depth data per environment, using a random policy. We apply domain randomization to textures of all objects, lighting color and direction, camera field-of-view, and camera position. For the best map to the canonical state, we train a Sim2Seg model per each environment and a separate Sim2Seg model for the combined dataset. During training, we use each environment's Sim2Seg model, which avoids distribution shifts between the Unity environments. At real-world test time, we also consider a Sim2Seg model trained on all environments to achieve optimal performance.

### 3.3   Sim2Seg Training

Following RCAN [1], we use conditional GANs (cGAN) with the U-Net Architecture [37, 31] to translate pairs of randomized simulation images into their canonical segmentation map representation. During train time, we use paired simulation data $(x_s, x_c, m_d, m_o)_{j\,j=1}^{N}$, where $x_s$ is a randomized RGB simulation image, $x_c$ is the canonicalized segmentation map, and $m_d$ is the canonicalized depth map, and $m_o$ the obstacle mask.

Unlike RCAN [1], we modify our objective to adapt to the task of autonomous driving. While RCAN predicts the canonical image, segmentation map, and depth map, we simplify our problem to predicting the segmentation map, which is simultaneously our canonical image; and the depth map, which is used as an auxiliary.

Thus, to encourage visual similarity, we optimize the following objective:

$$L_{eq}(G) = \mathbb{E}_{x_s,x_c,m_d,m_o}[\lambda_x l_{eq_x}(G_x(x_s), x_c) + \lambda_d l_{eq_d}(G_d(x_s), m_o \cdot m_d)]] \tag{1}$$

where $l_{eq_x}$ is the cross-entropy loss, $l_{eq_d}$ the L1 loss, $m_o \cdot m_d$ is the element-wise product, and $\lambda_x, \lambda_d$ the weighting of the losses. We denote $G_x(x_s)$ to be the segmentation output and $G_d(x_s)$ to be the depth output of randomized simulation image $x_s$.

**Adversarial Objective**   We employ a discriminator $D(x)$ which outputs the probability that the RGB image and segmentation map is a pair from the simulation dataset. Because our canonical output consists of probabilities for segmentation classes, we experimented with a few approaches to use $f$ to featurize the combination of RGB and segmentation map.

$$L_{GAN}(G, D) = \mathbb{E}_{x_s,x_c}[\log D(f(x_s, x_c)] + \mathbb{E}_{x_s}[\log(1 - D(x_s, f(G_x(x_s))] \tag{2}$$

Because our Sim2Seg model outputs segmentation class logits while the ground truth segmentation maps are one-hot encodings, discriminating directly on image and segmentation pairs is immediately trivial. In addition, because segmentation classes are discrete, we cannot sample a segmentation class and pass the gradient back to the generator. To remedy this issue, we initially attempted two two approaches: (1) sampling with gumbel-softmax [38] and (2) approximately the arg-max with soft arg-max [39], both of which are differentiable. Despite this, GAN training remained unstable, as discriminating between real and fake pairs was easy, likely due to noisy gumbel-softmax samples and soft-argmax values in areas of uncertainty.

Thus, to circumvent these issues, we choose instead to separately convolve the simulation image and segmentation maps to an equal number of channels to use as paired features for the discriminator to evaluate [40]. This allows for gradient flow without leading to unrealistic segmentation maps.

We find this to be effective for Sim2Seg, as it circumvents the problems of different modalities and instead allows for stabler discrimination in feature space, and we present ablations in Appendix E.1.

### 3.4 End-to-end Driving with RL

Using Sim2Seg, we train a short-horizon navigation policy using RL; specifically, we consider a goal-conditioned policy conditioned on visual and odometry data. Sim2Seg is inherently compatible with any visual learner, but we choose to use TD3 [41] with image augmentation [42] as our backbone RL algorithm. TD3 is an off-policy algorithm, and so enables goal-conditioning and relabelling in our training pipeline. We detail specific details about our architecture and hyperparameters in Appendix B.

**Observation Space**  We adjust the TD3 backbone policy with our augmented visual observations and state information for our domain. Before inference, Sim2Seg translates the input RGB image $o_t \in \mathbb{R}^{(3,256,256)}$ into a one-hot, $C$-class segmentation map $c_t \in \mathbb{R}^{(C,256,256)}$. Additionally, we condition on egocentric past trajectory $\tau_p \in \mathbb{R}^{10,3}$, the current 2D state $s_a \in \mathbb{R}^2$, and the goal state $s_g \in \mathbb{R}^2$. The encoded segmentation map and state information are flattened and concatenated to achieve a final representation.

**Action Space**  We parameterize the policy's actor as an LSTM which outputs a series of 5 action tuples, each consisting of a steering angle $\theta \in [\frac{\pi}{4}, -\frac{\pi}{4}]$, and acceleration $\alpha \in [0, 1)$. The policy then performs a temporal rollout of the actions to form a trajectory, which is consumed by a lower-level controller for vehicle commands. This parameterization comes with several benefits; proposing rollouts instead of vehicle torque commands mitigates the dynamics sim-to-real gap. Furthermore, proposing multiple continuous actions allows the policy to propose waypoints and consider more coherent short-term plans, while giving a notion of safety when executing in the real world. We perform an analysis of different action parameterizations in Appendix 5.3.

**Reward Function**  We design a simple reward function to best achieve goals while producing safe behaviors, notably obstacle avoidance:

$$r_t(s_g, s_a, a) = \lambda_g r_g(s_g, s_a) + \lambda_u r_u(s_g, s_a, a) + \lambda_s r_s(a) + \lambda_c r_c(s, a, s') \tag{3}$$

where $r_g$ is a goal-conditioned sparse reward, $r_u$ is an upright reward intended to incentivize smooth terrains (i.e. less rocky and flatter terrains), $r_s$ is a steer penalty intended to discourage bang-bang control, and $r_c$ is a collision penalty:

$$r_g(s_g, s_a) = \begin{cases} 100 & if \|s_g - s_a\|_2 < 2 \\ -1 & \text{otherwise} \end{cases} \qquad r_c(s, a, s') = \begin{cases} -1 & \text{if collision} \\ 0 & \text{otherwise} \end{cases}$$

$$r_u(s_g, s_a, a) = -\frac{|\theta|}{180} \qquad r_s(a_{\theta,\alpha}) = -\|\theta\|_2$$

where $\theta$ is the max of the roll and pitch angles between the vehicle body and world frames, and collisions are detected via Unity. To supplement training, we additionally leverage Hindsight Experience Replay, [43]: by relabeling sampled trajectories $(\tau, s_a, s_g)$ with a goal achieved later in the same trajectory $(\tau, s_a, s'_a)$ during training, we obtain more signal from the sparse reward $r_g$.

### 3.5 Real-world Vehicle Integration

For real-world evaluation, we use a Polaris S4 1000 Turbo RZR equipped with a variety of perception sensors, including an inertial measurement unit (IMU), stereo camera pairs, and LiDARs (see Figure 3). Note that we only use a monocular RGB camera when deploying. The Polaris also includes computing resources and a "drive by wire" system to autonomously control the vehicle (accelerate, change gears, brake, steer). NeBula [44] has been integrated with the vehicle and utilizes a ROS stack with planners for varying goal horizons. To integrate our

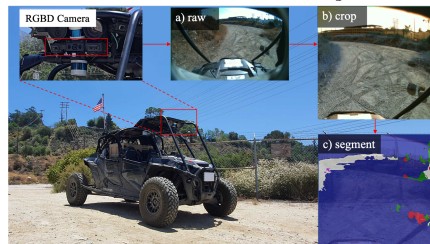

Figure 3: Polaris hardware with sensor suite. a) raw image, (b) cropped image as an input to Sim2Seg (c) segmented result using on-board computer.

model with the Nebula autonomy system, we further test in the ARL simulator, a photorealistic simulator of the Meadow environment already integrated with the ROS stack.

During real-world deployment, we plan in an iterative closed loop at 10 Hz, executing actions until the policy receives sufficient information. The policy stores a buffer of received messages, and only executes when timestamps are synchronized within 100ms. Output trajectories are consumed by a PID controller, which translates trajectories into low level vehicle commands.

# 4 Experimental Setup

**Preliminary: Classical Baseline** We construct the classical baseline by leveraging the the Nebula software stack [44]. Our specific implementation uses localization estimates from a LiDAR's inertial odometry to fuse depth scans temporally which are used to estimate traversability cost using a settling-based geometric analysis [45]. This traversability map is used by a kinodynamic motion planner [46] to generate collision-free trajectories that are followed by a lower level PID-based tracking controller.

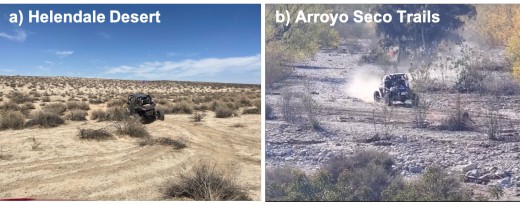

Figure 4: Real-world off-road evaluation data gathered by our platform at a) **Helendale**, Mojave Dessert and b) **Arroyo Seco Trails**, Altadena

**Offline Sim-to-Real Transfer Evaluation** To evaluate our policy's ability to perform visual sim-to-real transfer, we evaluate our policy using offline rosbag datasets of real-world, manual rollouts with the Polaris vehicle. Given rosbags of vehicle interactions, we first construct a dataset $\mathcal{D} := \{o, s_g, \tau_p^*; \tau^*\}$, where $o$ is the observation at $t = 0$, $s_g$ is the real achieved goal at some $t > 3$ seconds, $\tau_p^*$ is the past trajectory up to $t = 0$, and $\tau^*$ is the vehicle's real-world trajectory from $t = 0$. To calculate $\tau^*$ and $\tau_p^*$, we transform the recorded odometry into the vehicle's egocentric frame. We are then able to perform the model's full inference pipeline using $o$, $s_g$, and $\tau_p$ to produce a predicted trajectory $\tau$. Following this method, we construct two different real-world datasets from environments semantically and visually different from the set of training environments, which features the vehicle navigating winding trails, rocks, and vegetation (see Figure 4).

**Online Sim-to-Real Transfer Evaluation** We also perform qualitative evaluation of closed-loop control with our policy fully in the real world using the system described in Section 3.5 in the Arroyo environment.

# 5 Experimental Results

We seek to answer two primary questions: (1) Is Sim2Seg able to efficiently reach goals and perform obstacle avoidance during zero-shot transfer? (2) What factors are most necessary for Sim2Seg's performance?

## 5.1 Goal Reaching and Obstacle Avoidance

We evaluate the policy's capacity for obstacle navigation through comparison against real world rollout $\tau^*$, under the assumption that $\tau^*$ is an efficient trajectory to reach $s_g$ while avoiding obstacles. Following the offline evaluation setup described in Section 4, we define *efficient* as (1) reaching $s_g$ in a timely manner ($s_g$ is the offset in $t = 3$ seconds), and (2) avoiding obstacles (the human driver purposefully avoids collisions).

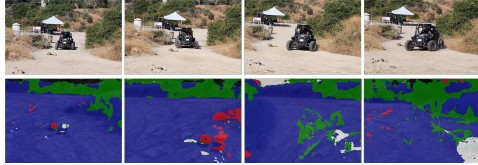

Figure 5: Timelapse of experimental demonstration of zero-shot transfer of goal following while avoiding obstacles.

We compare both the normalized L2 distance (denoted as **L2**) between the requested goal and the trajectory endpoint, and the angle difference between 10 evenly sampled points from $\tau^*$ and $\tau$ (denoted as **GT**) We also include absolute trajectory error (denoted as **ATE**), a classical measure of trajectory alignment. Additionally, to account for the possibility that $\tau$ reaches the goal more

directly than $\tau^*$, we also introduce $\mathbf{GT_G}$, which is the GT metric described above between the shortest distance to $s_g$ and $\tau$. A lower L2 and $GT_G$ indicates that the policy outputs trajectories that reach the goal (goal reaching); lower GT and ATE indicates that the policy aligns well with the human trajectory (obstacle avoidance).

To understand the gains provided by our method, we consider Sim2Seg's performance against several baselines detailed below. The results are listed in Table 1.

- Random: An untrained policy, i.e., random actions.
- Domain Randomization: We train an identical policy without the shared representation space learned through Sim2Seg, instead leveraging only our domain randomization methods.
- Classical: We consider an existing classical autonomous stack as described in Section 3.5.

| Data | Method | GT $\downarrow$ | ATE $\downarrow$ | $GT_G \downarrow$ | L2 $\downarrow$ |
|------|--------|------|------|------|------|
| Arroyo | Random | $0.354 \pm 0.002$ | $1.693 \pm 0.411$ | $0.343 \pm 0.005$ | $0.691 \pm 0.003$ |
| | DR | $0.292 \pm 0.026$ | $0.945 \pm 0.209$ | $0.280 \pm 0.022$ | $0.452 \pm 0.008$ |
| | Classical* | **0.070** | 0.812 | **0.087** | 0.653 |
| | Sim2Seg (ours) | $0.147 \pm 0.027$ | $\mathbf{0.471} \pm 0.012$ | $0.163 \pm 0.019$ | $\mathbf{0.287} \pm 0.024$ |
| Helendale | Random | $0.318 \pm 0.003$ | $2.128 \pm 0.029$ | $0.350 \pm 0.003$ | $0.751 \pm 0.005$ |
| | DR | $0.305 \pm 0.039$ | $1.546 \pm 0.374$ | $0.238 \pm 0.042$ | $0.501 \pm 0.028$ |
| | Classical* | **0.106** | 2.498 | **0.200** | 0.868 |
| | Sim2Seg (ours) | $0.158 \pm 0.007$ | $\mathbf{0.747} \pm 0.381$ | $0.210 \pm 0.013$ | $\mathbf{0.383} \pm 0.026$ |

Table 1: Offline evaluation metrics of our method against a broad set of baselines, including domain randomization (DR) and random actions (Random). Standard deviation shown across 5 seeds. *Note, Classical is not an RL solution, and so there are no seeds. Classical takes in LiDAR, whereas our other methods only takes in image and odometry inputs.

Furthermore, we demonstrate zero-shot transfer by deploying the algorithm in a real-world off-road environment on a passenger-size vehicle shown in Figure 5. In the included example, the policy identifies the rock as an obstacle, and creates a trajectory to navigate around it. Further videos can be found on the project webpage.

## 5.2 Ablation: Simulator and Real-World Data

We additionally compare the performance of our model against the sources of training data. Under *'1 Sim Env'*, we train only on one environment (Meadow), instead of all three simulator environments. Under *'Real-World'*, we additionally introduce a limited offline dataset of transitions, inspired by [1]. To construct this dataset, we invert the temporal rollout described in Section 3.4 on rosbags consisting of trajectories in a separate area of the Arroyo environment to obtain transitions $\{o, a, s_g, \tau_p^*; \}$, and add these to the replay buffer. During training, we sample 50% of our batch from the real-world transitions. Notably, this setup is amenable to using large amounts of offline, non-segmented data; RGB images are still segmented zero-shot according to a frozen pretrained Sim2Seg model. We experiment with different amounts of real-world data, including 0, 400, and 1200 transitions. Reducing the number of simulation environments doesn't seem to significantly harm model performance; we hypothesize that this could be in large part due to Meadow and Arroyo's similarities as relatively flat landscapes. However, introducing even as little as 400 transitions produces improved results and allows for even smoother transfer to the real world, shown in Table 2.

| Ablation | GT $\downarrow$ | ATE $\downarrow$ | $GT_G \downarrow$ | L2 $\downarrow$ |
|----------|------|------|------|------|
| Sim2Seg Baseline | $0.147 \pm 0.027$ | $0.471 \pm 0.012$ | $0.163 \pm 0.019$ | $0.287 \pm 0.024$ |
| 1 Sim Env | $0.189 \pm 0.020$ | $\mathbf{0.445} \pm 0.077$ | $0.174 \pm 0.021$ | $0.270 \pm 0.068$ |
| 400 Real World Transitions | $0.149 \pm 0.030$ | $0.489 \pm 0.057$ | $0.142 \pm 0.026$ | $\mathbf{0.266} \pm 0.153$ |
| 1200 Real World Transitions | $\mathbf{0.146} \pm 0.040$ | $0.448 \pm 0.156$ | $\mathbf{0.141} \pm 0.041$ | $0.432 \pm 0.060$ |
| Action = 1 Step | $0.309 \pm 0.135$ | $0.910 \pm 0.213$ | $0.234 \pm 0.060$ | $0.388 \pm 0.076$ |
| Action = 10 Steps | $0.158 \pm 0.045$ | $0.489 \pm 0.216$ | $0.180 \pm 0.046$ | $0.314 \pm 0.058$ |

Table 2: Incorporating real-world data is a scalable method to improve policy performance. Standard deviation shown across 5 seeds.

## 5.3 Ablation: Trajectory Parameterization

We additionally experiment with the number of actions $A$ our policy autoregressively outputs. $A = 1$ means the policy only takes one action at time; $A = 5$ means the policy, parameterized by an LSTM, generates 5 actions and takes 5 environment steps at a time. We evaluate the performance of a short-horizon policy ($A = 1$), medium-horizon policy ($A = 5$), and a long-horizon policy ($A = 10$). Our short-horizon policy performs noticeably worse, whereas our medium and short-horizon policies perform similarly well. This seems to suggest that learning longer-range behaviors can be important.

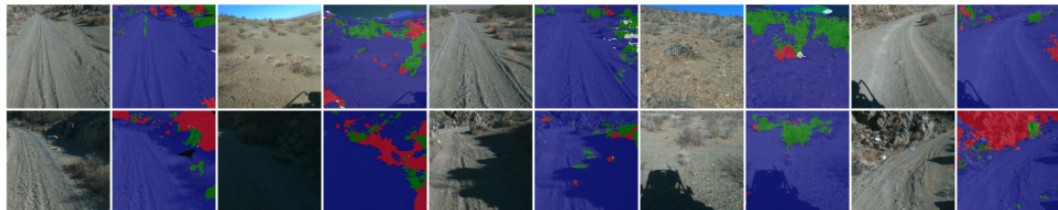

Figure 6: Qualitative zero-shot transfer results of our final Sim2Seg model. Our simulator environments range from a grassy meadow to a rocky canyon environment, yet with sufficient domain-randomization, we can achieve strong performance in the unseen and quite different real-world environments. Row 2 consists of observations we consider particularly challenging. Sim2Seg is able to generalize to harsh lighting and extreme shadows.

# 6 Conclusion and Limitations

In this paper, we have investigated transferring end-to-end off-road autonomous driving policies from simulation to reality. To accomplish this, we have presented Sim2Seg, which converts randomized simulated RGB images into segmentation masks, and subsequently enables real-world images to also be converted. Given that our driving policy is trained in these canonical segmentation environments, it is possible to run policies trained in simulation directly in the real world. When evaluating on real-world data, we are able to perform equally as well as a classical perception and control stack that took thousands of engineering hours over several months to build.

We differ from RCAN in that the discriminator on the visual features of the generated segmentation mask (rather than the canonical RGB image in RCAN). Given that our driving policy is trained in these canonical segmentation environments, it is possible to run policies trained in simulation directly in the real world. We show that not only is this a superior to training directly on domain randomization, but also that we are able to perform equally as well as a classical perception and control stack that took thousands of engineering hours over several months to build. To the best of our knowledge, this is the most ambitious visual sim-to-real transfer accomplished to date. We hope this inspires future sim-to-real work on even more ambitious domains.

**Limitations** Improving the quality of the Sim2Seg model is a key priority; qualitative analysis of segmentation maps shows difficulty with shadows (which may be perceived as new objects) and low-contrast scenes (such as the small shrubs depicted in Figure 6). We hope that such shortcomings can be addressed by introducing stronger shadow randomization techniques, and generally increasing the number of environments trained on.

Currently, Sim2Seg's policy is trained on horizons of 20 meters in length, and thus is only effective at this range. An effective horizon on the range of 50 meters or more would likely be much more practical, especially when navigating at higher speeds. Our policy currently is capable of avoiding obstacles in short range as demonstrated in Section 5.1; however, long horizon reasoning, such as needing to traverse around a forest to reach a goal, has not been tested. Training with temporally and spatially longer horizons inherently introduces more complexity to RL; this is an active area of research for us.

One particular area of interest is trajectory proposals in terms of waypoints. Waypoint proposals would allow us to explore other methods such as Bezier curve parameterizations and discrete trajectory libraries [47], and would provide further proof that Sim2Seg can generalize to different policy architectures.

**Acknowledgments**

This work was supported by DARPA RACER and Hong Kong Centre for Logistics Robotics. We would like to thank Valentin Ibars for his help running real-world experiments.

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
