# OpenReview forum: "Sim-to-Real via Sim-to-Seg: End-to-end Off-road Autonomous Driving Without Real Data"
_robot-learning.org/CoRL/2022/Conference — CoRL 2022 Poster_

### Official Review · Reviewer_ACNP · 2022-07-16

**Originality:** Good
**Technical Quality:** Very Good
**Clarity Of Presentation:** Very Good
**Impact:** 3

**Recommendation:**

Weak Accept: I recommend accepting the paper, but will not argue for my recommendation if the majority of other reviewers have a different opinion.

**Summary:**

The paper presents an autonomous end-to-end off-road driving approach which can be trained using only simulation data. To enable good sim2real transfer, segmentation maps are proposed as an intermediate representation and trained in an adversarial fashion inspired by RCAN. The segmentation maps can be easily generated in simulation and are shown to provide a suitable level of abstraction. The authors show experimental evidence that the method performs at least as good as a classical autonomous driving stack and that it can be deployed closed-loop on a real-world autonomous offroad vehicle.

**Issues:**

The authors should address the following issues to further strengthen their paper:
- Explain what a trajectory is and how the “GT” metric is computed.
- Motivate why the “GT” metric is a good choice compared to more classical metrics such as average trajectory error.
- Add details what the “efficient” trajectories executed by the human operator are and in what sense they are efficient.
- Explain why the difference to such an “efficient” trajectory is a good metric to evaluate the approach. Is it not possible, that the trajectory found by Sim2Seg is better?
- Discuss why adding the segmentation map intermediate representation helps overcome the issue that during training not all possible scenarios can be observed. Overall, the segmentation is also machine-learned and thus might suffer from the same problems as a direct end-to-end approach.


**

** AFTER REBUTTAL PERIOD **

**

The authors have addressed my comments in a satisfactory way: Specifically:
- _Explain what a trajectory is and how the “GT” metric is computed_: The explanations in the experimental section address this point fully.
- _Motivate why the “GT” metric is a good choice compared to more classical metrics such as average trajectory error_: As the authors have included the ATE as well in the revised version, this point is void. The ATE very clearly show that the proposed sim2seg method outperforms the baselines.
 - _Add details what the “efficient” trajectories executed by the human operator are and in what sense they are efficient.  Explain why the difference to such an “efficient” trajectory is a good metric to evaluate the approach. Is it not possible, that the trajectory found by Sim2Seg is better?_: The definition what efficient is (close to human driver), is still questionable but for the goal of this work, it makes sense.
- _Discuss why adding the segmentation map intermediate representation helps overcome the issue that during training not all possible scenarios can be observed. Overall, the segmentation is also machine-learned and thus might suffer from the same problems as a direct end-to-end approach._ The argument made by the authors that the segmentation map has much smaller dimension than the original image space is a good point. Furthermore, the authors have added ablations in section 5.4 which strengthen this argument.



**Quality Of The Limitations Section:**

Limitations are addressed clearly

**Reviewer Expertise:**

3: The reviewer is fairly confident that the evaluation is correct

**Robotics Focus:**

Sufficient demonstration on hardware

**Strengths And Weaknesses:**

The main strengths of the paper are its real-world experiments and evaluations that demonstrate the applicability of the proposed method. Using a true-to-scale demonstrator is a great contribution, in contrast to using only small RC cars or simulation only experiments.

On a more detailed level, the insights into what a suitable trajectory parametrization is for learning are interesting as they are not only specific to offroad driving but most likely a general finding. It would be interesting to see (as possible future work) if it is possible to design a neural controller that is directly capable of controlling the vehicle. Such a controller might inherently have advantages because it has access to all the information in the network and it thus might learn ‘careful’ driving when there is high uncertainty, or similar human-like behaviour.

The detailed description of the method in Sec. 3 is of great value to the reader and the community. It is well written and informs the reader about the key details of the proposed Sim2Seg approach.

However, since the experiments are a key part of this work, some clarifications might be required to further strengthen the results:
- It seems that the real-world rollouts collected for evaluation purposes are collected by having a human operator drive the vehicle around. It is stated that those generated trajectories are an “efficient” way to reach some specified goal. First, what does efficient mean in this context? Energy efficient? Time optimal? Second, since the reference for an “efficient” performance is generated by a human, there is no guarantee that there might not be an even more efficient solution, w.r.t to the efficiency-metric. That also implies that comparing the (possibly suboptimal) human trajectory with the results of the autonomous driving method might lead to a less accurate evaluation because the autonomous vehicle might be able to find a better trajectory.
- What are the ground truth trajectories $\tau$ and how are they recorded? Is a trajectory simply a list of positions associated with a time-stamp? Is it recorded via GPS only, RTK-GPS, GPS+IMU, ...
- In the evaluation (sec. 5.1) two metrics are used: L2 and GT, whereas the L2 metric only measures the difference between the trajectory endpoint and the specified goal. It is unclear what exactly the GT value characterizes. Why is cosine-similarity between two trajectories (assumed to be positions and 2D or 3D space) a good measure? Why is no classical measure like average trajectory error used for the evaluation? It would be very convincing to also provide the trajectory errors as well.

In the introduction and related work it is mentioned that a problem of RL methods is that it is very difficult to cover all possible environments, lighting cases, etc and that there are often “visual discrepancies”. The presented method learns the segmentation maps from directly from images. It would be very interesting to add details as to why the intermediate representation “segmentation map” is helpful to improve the robustness when unseen imaging conditions are observed.

Minor points:
- What low-level controller has been used to track the rollout?
- in line 300, there is a typo: “ultiple” instead of “multiple”
- Fig 3 is presumably done using wrapfig and the next page looks messed up.

**Summary Of Recommendation:**

The paper presents an approach that is based on RCAN work, but adapts the methodology to the new segmentation map intermediate representation. The work is well executed, although some details of the experimental evaluation need to be clarified, especially with respect to the choice of the optimal trajectory that the method is compared to. The inclusion of real-world experiments is really nicely done, especially considering the fact that the autonomous vehicles used in this work is true-to-scale.
At this point, I recommend "weak reject" because some parts of the experimental evaluation are not perfectly convincing. If the paper is improved a bit more, it will be a valuable contribution to the community.

---

> ### Author Response · Authors · 2022-08-22
> **Response to Reviewer ACNP (1/2)**
>
> We thank the reviewer for carefully reading the paper and providing constructive feedback. Here we will address any questions and concerns. In a few days, we will upload a new revision of our paper with additional experiments and an improved video.
>
> **1. Explain what a trajectory is and how the “GT” metric is computed. Motivate why the “GT” metric is a good choice compared to more classical metrics such as average trajectory error.**
>
> We define a trajectory as an arc of 10 points in $R^2$ constructed by rolling out actions as described in our supplementary materials (Section D Trajectory Rollouts). For the “GT” metric, we update the definition to instead be the angle difference between 10 evenly distanced sampled points along each spline. To motivate this choice, we considered the following factors: (1) penalize more trajectories that go overly straight when needing to turn, and (2) penalize less trajectories that go in the correct direction, but are overly short.
>
> However, the ATE (average trajectory metric) brought up is a great suggestion, and is a strong metric previously not considered. We have updated all tables in the paper revision with the ATE metric. Notably, in Table 1, Sim2Seg outperforms all baselines by a sizable margin in both the Arroyo and Helendale datasets.
>
> **2. Add details what the “efficient” trajectories executed by the human operator are and in what sense they are efficient.**
>
> We use “efficient” to attribute that (1) the human operator reaches the goal, and that (2) the human operator does not run into any obstacles while attempting to reach the goal. We can make these assumptions because of the evaluation setup; we set the “goal” of each trajectory to be 3 seconds into the future (meaning that the goal must be reachable within 3 seconds), and the human driver performs obstacle avoidance. We have made our definition of “efficient” more clear in the revision.
>
> **3. What are the ground truth trajectories τ and how are they recorded?**
>
> To collect these trajectories, we use odometry collected via GPS, IMU, and LIDAR. From the odometry, we use the world position of these points, and convert them to the vehicle’s frame during evaluation. Specifically, to collect the ground truth trajectory $\tau$, we consider a buffer of all odometry points collected in a 10 second interval. Upon selecting the initial time ($t=0s$), $\tau$ consists first of all recorded points up to $t=3s$, then interpolated to 10 evenly sampled points.
>
>
> **4. Explain why the difference to such an “efficient” trajectory is a good metric to evaluate the approach. Is it not possible, that the trajectory found by Sim2Seg is better?**
>
> We agree that the metrics we have found are not necessarily representative of “optimal” behavior. However, given real-world limitations (notably that testing real-world requires the presence of safety drivers) and abundant availability of real-world data, we reason that using such offline datasets presents a nice proxy of our algorithm’s capacity to cross the visual sim-to-real gap, which is an important aspect of our problem. The trajectory found by Sim2Seg may be better than the human trajectory (i.e. goes more directly to the goal), but it is difficult to quantify this; to attempt to do so, we have introduced in Table 1 a “goal GT” metric, which measures trajectory alignment to the shortest distance to the goal. Performances on "goal GT" are similar to standard GT for the Arroyo dataset between Classical and Sim2Seg (GT: 0.070 vs. 0.147; GT_g: 0.087 vs. 0.163) suggesting that the expert demonstrations are fairly aligned; however, on the Helendale dataset, the gap shrinks considerably (GT: 0.106 vs. 0.158; GT_g: 0.200 vs. 0.210), suggesting that the expert trajectories are somewhat misaligned. Following the reviewer's advice, we include the GT_g to capture this gap in the offline dataset.

---

> > ### Author Response · Authors · 2022-08-22
> > **Response to Reviewer ACNP (2/2)**
> >
> > **5. Discuss why adding the segmentation map intermediate representation helps overcome the issue that during training not all possible scenarios can be observed. Overall, the segmentation is also machine-learned and thus might suffer from the same problems as a direct end-to-end approach.**
> >
> > There is indeed a visual gap in segmentation maps; however, we believe using the intermediate representation of segmentation maps is helpful in improving robustness and bridging the domain gap. RGB images, for example, are extremely high dimensional (256x256x256x3 = ~50 million) whereas segmentation images are sparse one-hot representations and relatively lower dimensional (256x256xnum_classes = 256x256x6 = ~400k). Thus, the distribution shift in image space has the potential to be large, whereas we may lessen the issue of distribution shifts by using a more compact state.
> >
> > To validate this hypothesis, in Appendix E of our original submission (and Table 2 of our revision), we included comparison of pure domain randomization in image space and our Sim2Seg model. Though both methods rely on domain randomization, in real-world evaluations, our domain-randomization agent, which learns to be robust to visual RGB inputs performs worse than our Sim2Seg model, which learns to act with segmentation inputs. This suggests that while there may be a domain gap in both, our agent is much more robust when given segmentation maps than RGB images. We will also include updated tables in our revised submission.
> > We acknowledge that there is still a domain gap in the segmentation image space, and we introduce additional experiments to bridge this gap. We introduce additional experiments with real-world data mixing during agent training so that our agent can be robust to real-world Sim2Seg outputs, which improves our performance.
> >
> > **5. What low-level controller has been used to track the rollout?**
> >
> > We use the same PID controller as described in Preliminary: Classical Baseline.
> >
> > Our proposed sim2real model generates an off-road obstacle avoidance path. To stay close to this path, MPC generates throttle, position and velocity commands which is 5 second long at about 30Hz. PID tracks this trajectory at 100Hz.

---

### Official Review · Reviewer_aEoJ · 2022-07-30

**Originality:** Very Good
**Technical Quality:** Very Good
**Clarity Of Presentation:** Excellent
**Impact:** 4

**Recommendation:**

Strong Accept: I recommend accepting the paper and will argue for my recommendation even if other reviewers hold a different opinion.

**Summary:**


The paper proposes sim2seg, which aims to surpass the sim-to-real gap for off-road autonomous driving without using real-world data. The proposed method converts simulated RGB images of terrain to segmentation and depth masks and subsequently enables converting real-world images. The segmented images are then used to train a navigation policy using reinforcement learning. The proposed approach is proven efficient in reaching goals and performs obstacle avoidance during zero-shot transfer. The paper also investigates the various factors necessary for sim2seg’s performance.

**Issues:**

+ I would request the authors provide more details of their approach in the Introduction section.
+ Please check the indentation in line 207.
+ Please include details on the method's runtime, how the approach can adapt when sensor data is obtained at lesser frame rates, and if the approach is robust to the various lighting conditions.



**Quality Of The Limitations Section:**

Limitations are addressed clearly

**Reviewer Expertise:**

5: The reviewer is absolutely certain that the evaluation is correct and very familiar with the relevant literature

**Robotics Focus:**

Sufficient demonstration on hardware

**Strengths And Weaknesses:**

Strengths:

+ The paper provides a clean and straightforward method for sim-to-real learning for off-road autonomous driving.
+ The paper is well written and easy to follow for a broad robotics audience.
+ The proposed method is evaluated on a physical robot platform. It appears to cross the sim-to-real visual gap successfully.
+ The authors have comprehensively detailed the approach's limitations, providing insights into the future direction of this research.


Weakness:

+ It will be helpful to include more details of the approach in the Introduction section.
+ It might also be helpful to understand the approach from a high level by including a diagram describing the various modules of the proposed approach.


**Summary Of Recommendation:**

The paper presents a method for solving the challenging sim-to-real transfer problem without using any real-data. The paper is well written and easy to follow. The experimental validation on the physical hardware platform makes the paper a good contribution.

---

> ### Author Response · Authors · 2022-08-22
> **Response to Reviewer aEoJ**
>
> We thank the reviewer for carefully reading the paper and providing constructive feedback. Here we will address any questions and concerns. In a few days, we will upload a new revision of our paper with additional experiments and an improved video.
>
> **1. It might also be helpful to understand the approach from a high level by including a diagram describing the various modules of the proposed approach.**
>
> Agree that this would indeed aid understanding. We attach a revised high level diagram; let us know if this provides helpful insight.
>
> **2. Please include details on the method's runtime, how the approach can adapt when sensor data is obtained at lesser frame rates, and if the approach is robust to the various lighting conditions.**
>
> During deployment, we use our planner iteratively at a maximum rate of 10Hz, executing actions until the planner receives sufficient information to replan. To adapt to sensor data frequencies, we only plan when RGB image and past world odometry timestamps are synchronized within 100ms. Specifically, the past trajectory is manually calculated from odometry, using a running buffer collected in parallel with serving transformed trajectories. The goal state is stored in reference to the world frame, and transformed during planning. Similarly, the RGB image is segmented during planning. We have updated the “Real-World Vehicle Integration” section with these details in the revision.
>
> Regarding various lighting conditions, we deploy lighting and shadow randomization during segmentation model training in an attempt to learn invariance to such lighting conditions. Qualitatively, we demonstrate performance in the same environment with differing lighting conditions -- this is in the updated paper. The agent seems to generalize well despite the change in lighting; we can attribute this to the learned invariance in the segmentation map.

---

### Official Review · Reviewer_McEn · 2022-07-31

**Originality:** Fair
**Technical Quality:** Fair
**Clarity Of Presentation:** Very Good
**Impact:** 2

**Recommendation:**

Weak Reject: I recommend rejecting the paper, but will not argue for my recommendation if the majority of other reviewers have a different opinion.

**Summary:**

This paper addresses the problem of developing an
end-to-end controller for autonomous driving. In
particular, the authors focus on training a
controller in simulation (where no real data is
required) and then bridging the sim-to-real gap
via a segmentation network (also trained on
simulation data). Evaluation is provided in both
an online and offline setting.

**Issues:**

1. I would like to see more justification for the proposed image convolution approach to training the GAN. In particular, Wassertein GANs are known to overcome many of the GAN training stability issues, so that would be one alternative approach to compare with. For example, the authors can look at this paper and code within: https://proceedings.neurips.cc/paper/2017/file/892c3b1c6dccd52936e27cbd0ff683d6-Paper.pdf

2. The authors are encourage to discuss in greater detail the differences with RCAN and/or provide an experimental comparison.

**Quality Of The Limitations Section:**

Limitations are not well addressed

**Reviewer Expertise:**

4: The reviewer is confident but not absolutely certain that the evaluation is correct

**Robotics Focus:**

Sufficient demonstration on hardware

**Strengths And Weaknesses:**

Overall, the paper is well written and the problem
is well motivated. Bridging the sim-to-real gap
would enable the development of controllers that
do not require real data and yet are exhaustively
tested in simulation. I also appreciate the
comparison between the proposed system and the
classical software stack approach. However, I do
have a few comments that can be used to improve
the paper's quality.

My first comment has to do with the comparison
with the classical approach. The authors make a
very strong claim that "our trained RL agent can
perform as well as a sophisticated, model-based
autonomous driving stack". However, the comparison
is performed in a very controlled setting, with
short tasks, where the goal is known, and in
environments that are likely too similar to the
simulation environment and hence not exhibiting
significant distribution shift. In order to
justify this claim, I think the authors need to
illustrate cases where the proposed method fails,
especially in comparison with the classical
method. Of course, experiments where the classical
method fails would be equally interesting.

Furthermore, the contribution over the RCAN method
seems unclear to me. It seems that the proposed
method is almost identical to RCAN in terms of
the conversion to a "canonical" representation of
each observation, the main difference being that
the controller is trained using TD3. In my opinion,
this is not a major contribution. If the authors
focus more on the off-road application, perhaps
that would separate it more from the RCAN work.

Related to the above, it is not clear to me that
the GAN training adjustment (i.e., the image
convolution) is needed. In particular, the original
GAN training algorithm is known to be unstable,
and multiple improvements have been proposed, e.g.,
Wasserstein GANs. The authors are encouraged to
explore this issue further.

Minor:

I do not understand why the past trajectory is in
R^{10,3}. If I understand correctly, the trajectory
consists of 10 time steps, but I do not get where
the 3 comes from -- it seems that state is in R^2.

I also do not see why the action space consists
of 5 action tuples. Are these 5 tuples predictions?
If so, the authors are encouraged to comment on this
design choice, as well as on the significance on
the number 5.

A lower-level comment I have is that multiple
symbols in the paper are not properly explained.
For example, L_eq equation on page 4 has several
unintroduced terms, e.g., G_x, G_d, etc.

**Summary Of Recommendation:**

I am recommending a weak reject because I don't think the contributions are significant enough. In particular, it seems that the improvements over RCAN are minor and that the GAN training improvements are not justified through experiments and comparisons.

---

> ### Author Response · Authors · 2022-08-22
> **Response to Review McEn (1/2)**
>
> We thank the reviewer for carefully reading the paper and providing constructive feedback. Here we will address any questions and concerns. In a few days, we will upload a new revision of our paper with additional experiments and an improved video.
>
> **1. The sim2real gap is evaluated in a very controlled setting, with short tasks, where the goal is known, and in environments that are likely too similar to the simulation environment, and hence not exhibiting significant distribution shift.**
>
> Thanks for the comment. We have now added more details on this front to the paper to clarify the distributional shift. In particular, we want to highlight the difference between previous works and our off-road driving application, a much more challenging setting. RCAN presents experiments in robotic manipulation in controlled lab settings, where the camera is fixed, and the camera view is controlled (pointed towards a table) to only contain objects that, even if unseen, are relatively similar. Our camera view, in contrast, is truly in the wild. We have almost no control over what the camera will see during test time, with no guarantees of geometric, visual, or even perceptual similarity between the real-world environments and the simulation environments. Frequent changes in the elevation under any of the vehicle wheels in offroad driving causes significant changes in the sensor readings, image quality, and field of views. Ignoring the noted sim-to-real visual gap between even geometrically similar environments, our training environments vary drastically from the chosen testing environment, both semantically (in terms of class distribution) and visually. “Meadow” sim is a lush green field with a forest, “Landscapes” sim is populated with trees, and “Canyon” sim primarily features rocky outcrops. Our real-world evaluations include two diverse settings that are not only different from each other but the simulators: Helendale, a desert-like environment with small bushes, rocks, and hills; and Arroyo, a dusty path with dense trees. Additionally, we never explicitly train on trail-following tasks, and only reward obstacle avoidance.
>
> To further illustrate, we have now updated our website with visual examples from both the training environments, and the testing environments. Please reference the Rebuttals page under the heading “Reviewer McEN”: https://sites.google.com/view/sim2segcorl2022/home/rebuttal-materials
>
> **2. Seems that the proposed method is almost identical to RCAN in terms of the conversion to a "canonical" representation of each observation, the main difference being that the controller is trained using TD3. If the authors focus more on the off-road application, perhaps that would separate it more from the RCAN work.**
>
> We agree that the main contribution of this paper is its novel zero-shot application in off-road terrains, which presents new challenges in sim2real. However, as we discovered, the devil is in the detail as to *what* canonical representation is given to the RL agent, segmentation maps were chosen for this work, while RCAN used RGB images as the representation. In Table 2, we present experiments with an RCAN intermediate representation (see the response below for more information); however, it should be noted that for such diverse environments as off-road driving, RCAN outputs are notably worse than Sim2Seg outputs, and learning in such a canonical state is actually ineffective.
>
> We comment above on the difficulties of bridging the sim-to-real gap; in particular, our experimental setting differs from robotic grasping or manipulation in the drastic variation in the environment. Our agent navigates in uneven, rocky terrain, leading to constant changes in the diverse environment. Obstacles like shrubs, rocks, boulders, and trees can vary in appearance, shape, size, and density based on the geographic location.
>
> In addition, while we would like to highlight that Sim2Seg’s performance should be agnostic of the chosen algorithm, much care was taken in designing this specific controller; we present our work as a comprehensive pipeline, including the simulations and action modes, both of which present some novelty.

---

> > ### Author Response · Authors · 2022-08-22
> > **Response to Reviewer McEn (2/2)**
> >
> > **3. The authors are encouraged to discuss in greater detail the differences with RCAN and/or provide an experimental comparison.
> > Below, we highlight the major differences:**
> >
> > Thank you for the comment. We have now explicitly discussed these differences in the Related Works section of the revised version. Below, we recap the major differences:
> >
> > 1. RCAN converts visual inputs to canonicalized RGB images. Sim2Seg converts visual inputs to segmentation maps, which are a different modality. The policy trained in RCAN acts in image space, while the Sim2Seg policy acts upon segmentation maps.
> > 2. RCAN discriminators take in paired RGB images. To improve discriminator stability, our discriminator takes in learned features of input images and segmentation maps.
> > 3. RCAN shows experiments in robotic manipulation. Sim2Seg evaluates autonomous driving in the much more difficult environment of offroad environments, where there are greater visual challenges (lighting conditions, huge variation in types and distribution of obstacles).
> >
> > To address (1), in Table 2, we have included additional comparisons to the revised version, where we use the RCAN model to convert visual inputs to canonicalized RGB images. These differences have been made more explicit in the revision.
> >
> > **4. I do not understand why the past trajectory is in $R^{10,3}$. If I understand correctly, the trajectory consists of 10 time steps, but I do not get where the 3 comes from -- it seems that state is in $R^2$.**
> >
> > Great point: this has now been made clear in the revision. All data is actually captured in $R^3$, but we preprocess the goals to be $R^2$ during deployment and training; we care only about the flat coordinate requested. Keeping the odometry / past trajectory as $R^{10\times 3}$ was a design decision, with the rationale that it could potentially be useful to capture nuances such as the bumpiness in the trajectory. This has been made clear in the revision.
> >
> > **5. Why does the action mode consist of 5 action tuples, and what is the significance of the number 5?**
> >
> > Via running experiments with 1, 5, and 10 autoregressive actions, we chose to autoregressively output 5 actions as a design choice. This appears in Table 2 of the revision.
> >
> > **6. I would like to see more justification for the proposed image convolution approach to training the GAN. In particular, Wassertein GANs are known to overcome many of the GAN training stability issues, so that would be one alternative approach to compare with.**
> >
> > Comparing to Wassertein GANs is a great suggestion, and in Table 2, we present results from the experiments with the WGAN penalty loss in order to see how different GAN formulations may affect the quality of our segmentation maps, and ultimately, the performance of our agents trained on these segmentation maps.
> >
> > We proposed our image convolution to bypass a couple of issues.
> >
> > (1) Our segmentation classes are one-hot logits, and convolving to feature space avoids sampling differentiably. We experiment with both soft arg-max and gumbel soft-max to sample differentiably, but we find that resulting samples are unrealistic, making it trivial for the discriminator to discriminate between real and fake segmentation maps. By convolving to feature space, we avoid such instabilities.
> >
> > (2) Our segmentation maps are a different modality, and with a large number of classes, it may be unscalable and unbalanced to feed in the concatenated image and segmentation map to the discriminator. Though we only define 6 segmentation classes in our paper, future settings may need many more, even hundreds, of classes. Not only does this necessitate large discriminator architectures, but it also puts a huge constraint on the discriminator to identify relevant features from both domains. Discriminating in the feature-space bypasses this issue.
> >
> > (3) Empirically, convolving both RGB images and segmentation maps improves stability and performance. We compare our results with new experiments using the WGAN penalty in Table 2 to verify that this holds true.

---

### Official Review · Reviewer_Vqbb · 2022-08-05

**Originality:** Fair
**Technical Quality:** Good
**Clarity Of Presentation:** Good
**Impact:** 3

**Recommendation:**

Weak Reject: I recommend rejecting the paper, but will not argue for my recommendation if the majority of other reviewers have a different opinion.

**Summary:**

Instead of building a module-based ADS, this work aims to achieve off-road self-driving end-to-end. Concretely, a driving policy trained in a simulator can be directly transferred into the real world. The core technique enabling zero-shot transfer is a module built on RCAN, which can bridge the gap between the images captured in the real world and those in the simulator. Therefore, this method can avoid the dangerous data collection process of trial and error. Also, this method can be trained efficiently. With this method, the policy trained in 48 hours is comparable to a rule-based policy, which requires a huge human budget.

**Issues:**

N/A

**Quality Of The Limitations Section:**

Additional details required

**Reviewer Expertise:**

4: The reviewer is confident but not absolutely certain that the evaluation is correct

**Robotics Focus:**

Sufficient demonstration on hardware

**Strengths And Weaknesses:**

**Strength**:
1) This work builds a complete pipeline to show that training in a simulator and transferring the learned policy to the real world is feasible
2) The idea that mapping RGB images from different domains (simulation/reality) to the segmentation domain is interesting. This also enables the zero-shot transfer.
3) It is exciting to see the real vehicle experiment since it is difficult to do hardware experiments and a lot of researchers only validate their idea in simulators.

**Weakness**
1) The originality of the method is limited. It seems that this work transfers RCAN to the driving task with a small modification.
2) The standard deviation is missing in all tables. If repeated experiments are conducted, please report std.
3) The real-world demonstrative video is not striking enough. It shows unsmooth trajectories, and the setting is simple. Actually, learning to drive via RL is achieved several years ago: https://wayve.ai/blog/learning-to-drive-in-a-day-with-reinforcement-learning/.  This video is more attractive than the demo in the supplementary material. In my opinion, the core contribution of this work is to build such a hardware system and demonstrate that zero-shot sim-to-real autonomous driving can be completed. Therefore, the quality of trajectories in demo video is the most important part. If smooth and long driving trajectories or more complex scenarios can be shown, I will consider increasing my score.


typo:
line 156-157: two two approaches

**Summary Of Recommendation:**

The method is an incremental work built on RCAN, and thus originality is limited. Although hardware experiments are conducted, the driving trajectories are not of high quality.

---

> ### Author Response · Authors · 2022-08-22
> **Response to Reviewer Vqbb**
>
> We thank the reviewer for carefully reading the paper and providing constructive feedback. Here we will address any questions and concerns. In a few days, we will upload a new revision of our paper with additional experiments and an improved video.
>
> **1. The originality of the method is limited. It seems that this work transfers RCAN to the driving task with a small modification.**
>
> Thanks for the comment. We would like to emphasize that while our method is inspired by Sim-to-Real techniques from robotic manipulation, our application is novel and significantly different in the scale and the nature of the data, which leads to new insights and shifts in our method. There are many significant challenges that arise in off-road driving, and our method is the first, to our knowledge, to show end-to-end sim-to-real transfer, not only for self-driving, but in a vastly more visually complex offroad scenario than in previous work. RCAN presents experiments in robotic manipulation in controlled lab settings, where the camera is fixed, and objects are relatively similar, even if unseen. In particular, the vast and deep sensor field of views (with high variation of elements in a single image) makes the nature of offroad data significantly different from that of manipulation tasks. In addition, off-road terrains are much more visually complex than RCAN's robotic grasping setup, as these outdoor observations may change depending on weather, location, and lighting (much more than the typical lighting and color variation in grasping setups). Types of obstacles can range from pebbles, small shrubs, mud, large trees, and boulders, among many others, and both the distribution and types of obstacles may vary greatly depending on the location.
>
> **2. The standard deviation is missing in all tables. If repeated experiments are conducted, please report std.**
>
> We have updated all tables with updated numbers (mean, std) from 5 trials, and will appear in the revision.
>
> **3. Learning to drive via RL was achieved several years ago in https://wayve.ai/blog/learning-to-drive-in-a-day-with-reinforcement-learning/.**
>
> Thank you for highlighting this paper. Here, we discuss some of the main differences in problem statements between our work and that of Kendall et al (the blog post mentioned). We train a goal-conditioned policy; Kendall et al maximize distance without intervention, which means they are unable to specify a course for the vehicle; they note this as a limitation in their work. We also do not focus explicitly on lane following, but only off-road obstacle avoidance during training; thus, our real-world evaluation also presents a challenging shift in the task, as the vehicle is never explicitly exposed to “roads” during training. The main difference is that we transfer our policy zero-shot to real-world navigation, use photo-realistic simulation for transfer, and are compatible with offline, real-world data. On the other hand, to close the noticeable visual gap between simulation and reality in their method, Kendall et al requires real-world finetuning with the physical vehicle (which presents potential safety concerns, especially in large vehicles such as ours). Our addition of a domain transfer approach (segmentation maps) is also a noted line of future work from Kendall et al. We have updated our related work section to discuss this difference in approach.
>
> **4. Real world videos were not striking enough.**
>
> Thank you for the comment. We have now improved our videos and improved the method leading to smoother longer-range real-world trajectories.

---

### Author Response · Authors · 2022-08-26
**Paper Revision**

Attached is a revision of the paper, with all revisions/additions to the original text marked in blue. We highlight some of the larger changes:
1. In **Related Work**, we further highlight the differences between our method and works brought up by the reviewers (Kendall et al, James et al).
2. In **Real-world Vehicle Integration**, we add details regarding planning rates and sensor data.
3. In **Experimental Results**, we consolidate our results into two tables (baselines and ablations). To address reviewer concerns regarding metrics, we report the mean and standard deviation across 5 seeds, and introduce Absolute Trajectory Error (ATE). We additionally update our "GT" metric, and introduce another metric "GT_G" to account for the possibility that the proposed trajectory reaches the goal more efficiently than the human driver.
3. In **Experimental Results**, we further elaborate on what a trajectory is, how they are collected, and what an "efficient" trajectory is in evaluation.
4. In **Experimental Results**, we provide additional experiments using real-world data during agent training, presenting a scalable way to improve real-world transfer.
5. In **Experimental Results**, to address reviewer concerns regarding our proposed GAN improvements, we add experiments comparing downstream results when using our method, RCAN (i.e. Pix2Pix), and WGAN.
6. We add additional figures showcasing the visual and semantic domain shift present between the training simulation environments and real-world evaluation environments.

We have additionally uploaded videos of longer, improved real-world rollouts to the website, which can be seen here: https://sites.google.com/view/sim2segcorl2022/home/rebuttal-materials

---

### Meta-Review · Area_Chair_VdSs · 2022-08-15

**Recommendation:** Accept (Poster)
**Confidence:** 4

**Metareview:**

The paper is well written and the addressed problem is well motivated. The approach of using the segmentation map as an intermediate representation to solve the sim2real problem is interesting and the evaluation on a real, true-to-scale demonstrator vehicle was seen as a strong point and valuable contribution by all reviewers. In contrast, there was much more of a divided assessment regarding the originality of the approach and the significance of the contributions. In particular, the authors should address the concerns of reviewers McEn and Vqbb that the presented method seems to be a rather straightforward transfer of RCAN to the autonomous driving domain, and that progress over the original RCAN method remains unclear.

Other issues to address:
* No standard deviation is given in any of the results tables
* An illustration of failure cases of both the proposed method the classical method would be helpful to strengthen claims
* It is unclear that the proposed GAN training adjustment is really justified, especially in light of other GAN modifications that address training stability such as Wasserstein GANs
* More details on the method's runtime, robustness to slower sensor sampling frequencies to various lighting conditions would be helpful
* More details on trajectory definition and metrics used for evaluation would be appreciated
* Please discuss why adding the segmentation map intermediate representation helps to overcome the issue that not all possible scenarios can be observed during training, given that the map is itself learned from data

===

Post rebuttal/discussion update:

Thank you for providing a detailed rebuttal. In the internal discussion, reviewers agreed that their concerns were addressed properly and expressed the feeling that the paper adds value to the community.

---

> ### Author Response · Authors · 2022-08-28
> **Response to Area Chair VdSs (1/2)**
>
> We thank the AC for summarizing the crucial points made by the reviewers. We address each of the AC points directly below.
>
> **1. No standard deviation is given in any of the results tables**
>
> We have updated all tables with updated numbers (mean, std) from 5 trials, and will appear in the revision.
>
> **2. An illustration of failure cases of both the proposed method and the classical method would be helpful to strengthen claims**
>
> Our method relies primarily on learned visual inputs, so we may fail in settings where depth is hard to estimate (such as under certain lighting conditions including low-contrast settings) or where we encounter unseen obstacles in training (such as fences and other vehicles as demonstrated in the additional video on the website). In deployment, factors such as dirt can obstruct the camera view and introduce noise to our visual observations.
>
> The classical pipeline relies on several intermediate representations such as temporally fused maps constructed from odometry data which are then processed by the planner, while this decoupling makes the problem more tractable; it comes at the expense of increased sensitivity to odometry failures such as drift. Degraded sensing, such as mud/dirt covering part of the sensor field of view, mechanical displacement affecting extrinsic calibration, and similar sensor degradation have caused some failures in our testing.
>
> We do not yet have a lot of data with degraded sensor driving, but aim at collecting more data on this front in the future.
>
> **3. It is unclear that the proposed GAN training adjustment is really justified, especially in light of other GAN modifications that address training stability such as Wasserstein GANs**
> The primary reason for this contribution is because segmentation classes are discrete, and we wanted to avoid instabilities encountered with differentiable discrete sampling. Discriminating in the feature space of images and segmentation maps both improves stability and avoids the issue of sampling from discrete sampling classes. Empirically, we also find that convolving both RGB images and segmentation maps improves stability and performance downstream (see updated Table 2). We have added some additional details in our response to reviewer McEn.
>
> **4. More details on the method's runtime, robustness to slower sensor sampling frequencies to various lighting conditions would be helpful**
>
> Thanks for this comment. We have now included details regarding real-world deployment in our response to reviewer aEoJ.
> Regarding various lighting conditions, we add lighting and shadow randomization during segmentation model training in an attempt to learn invariance to such lighting conditions. We have also updated our paper with offline results from experiments conducted at different times of the day to evaluate how well our agent generalizes to real-world changes in lighting.
>
> We have included additional details in our response to reviewer aEoJ.
>
> **5. More details on trajectory definition and metrics used for the evaluation would be appreciated**
>
> Thank you for this comment. We have now added these details to the paper.
>
> We define a trajectory as an arc of 10 points in R^2 constructed by rolling out actions as described in our supplementary materials (Section D Trajectory Rollouts).
>
> For the “GT” metric, we have updated the definition to instead be the angle difference between 10 evenly distanced sampled points along the demonstration spline and our policy roll-out spline. The “GT” metric evaluates how close we are to an offline human trajectory, motivated by the fact that human drivers, more often than not, take reasonable routes to goals.
>
> Our “L2” metric is the L2 distance between the final endpoint of the policy rollout to the requested goal, in the case that our policy actually outperforms the human trajectory and achieves a closer position to the goal. We additionally introduce “GT_G”, which is the above GT metric but between the rollout and the shortest path to the goal, to further capture an “efficient trajectory” which reaches the goal by a shorter means than the human trajectory.
>
> The ATE (absolute trajectory metric) brought up in the review is a strong metric previously not considered. We have updated all tables in the paper to include the ATE metric.
>
> In our responses below, we have addressed the additional helpful feedback about trajectories and offline evaluation provided by reviewers McEn, ACNP. Our updated paper now includes clearer definitions and explanations of trajectories and evaluation.

---

> > ### Author Response · Authors · 2022-08-28
> > **Response to Area Chair VdSs (2/2)**
> >
> > **6. Please discuss why adding the segmentation map intermediate representation helps to overcome the issue that not all possible scenarios can be observed during training, given that the map is itself learned from data**
> >
> > There is indeed a visual gap in segmentation maps; however, using the intermediate representation of segmentation maps is a critical factor  in improving robustness and bridging the domain gap.
> >
> > To verify this insight empirically, we have experimental results that rely on pure domain randomization in image space. These perform considerably worse than our Sim2Seg policies, which take in segmentation maps. We also present results in our updated paper on the effect of incorporating real-world data: this greatly improves results, suggesting that seeing real-world Sim2Seg outputs improves policy performance. Thus, while there is a visual gap in segmentation maps, we alleviate this through using a small set of real-world data and avoid the larger (and harder-to-capture)  visual gap between simulated and real-world RGB images.